# Challenges to Antimicrobial Stewardship in the Countries of the Arab League: Concerns of Worsening Resistance during the COVID-19 Pandemic and Proposed Solutions

**DOI:** 10.3390/antibiotics10111320

**Published:** 2021-10-29

**Authors:** Nesrine A. Rizk, Rima Moghnieh, Nisrine Haddad, Marie-Claire Rebeiz, Rony M. Zeenny, Joya-Rita Hindy, Gabriella Orlando, Souha S. Kanj

**Affiliations:** 1Division of Infectious Diseases, Internal Medicine Department, American University of Beirut Medical Center, Beirut 1107 2020, Lebanon; nr00@aub.edu.lb; 2Division of Infectious Diseases, Department of Internal Medicine, Makassed General Hospital, Beirut P.O. Box 11-6301, Lebanon; moghniehrima@gmail.com; 3Pharmacy Department, American University of Beirut Medical Center, Beirut 1107 2020, Lebanon; nh126@aub.edu.lb (N.H.); rz37@aub.edu.lb (R.M.Z.); 4Faculty of Health Sciences, American University of Beirut, Beirut 1107 2020, Lebanon; mr88@aub.edu.lb; 5Division of Infectious Diseases, Internal Medicine Department, Mayo Clinic, Rochester, MN 55902, USA; joyahindy@gmail.com; 6Infectious Disease Clinic, Policlinico University Hospital, 41122 Modena, Italy; orlando.gabriella@aou.mo.it

**Keywords:** antimicrobial stewardship, antimicrobial resistance, COVID-19, Arab countries

## Abstract

The COVID-19 pandemic is expected to worsen the global problem of antimicrobial resistance (AMR). There is a heightened interest in understanding this effect and to develop antimicrobial stewardship (AMS) interventions accordingly to curb this threat. Our paper aims to evaluate the potential magnitude of COVID-19 on AMR and AMS with a focus on the countries of the Arab league, given the social, political, and economic environments. We also evaluate obstacles in applying the rational use of antibiotics, monitoring resistance trends in the midst of the pandemic, and evaluating the impact of the economic crisis in some countries. We aim to raise awareness about the potential effects of antibiotic overuse during the pandemic and to propose practical approaches to tackle this issue.

## 1. Introduction

The severe acute respiratory syndrome coronavirus 2 (SARS-CoV-2) is a novel human coronavirus that emerged in the city of Wuhan, China in December 2019 [1,2]. It rapidly spread worldwide causing the coronavirus disease 2019 (COVID-19) pandemic, as declared by the World Health Organization (WHO) in March 2020 [1,2,3]. This pandemic has already caused detrimental effects including major losses of human life, economic repercussions, and serious consequences for global health systems.

Antimicrobial resistance (AMR) has been recognized as a major threat to public health, economy, and security [4,5,6] at local and international levels [7]; it has spread to every continent and country in the world [8] and it is estimated that a continued rise in AMR would lead to 10 million deaths yearly across the globe by 2050 [9]. During the last decade, there was a dramatic rise in AMR, especially among Gram-negative bacteria [10,11]. Of particular interest is the emergence of multiple resistant strains including the ESKAPE organisms: *Enterococcus faecium*, *Staphylococcus aureus*, *Clostridium difficile*, *Acinetobacter baumannii*, *Pseudomonas aeruginosa*, Enterobacterales (*Klebsiella pneumoniae*, *Enterobacter* spp., and other resistant species including *Escherichia coli* and *Proteus* spp.). The ESKAPE pathogens are the leading cause of nosocomial infections throughout the world [12]. More recently, the WHO has classified the pathogens of concern into three categories: critical, high, and medium priority [13] based on morbidity, mortality, availability of treatment options, and drugs in the development pipeline. Of note, third generation cephalosporin resistant (3GCR) Enterobacterales and carbapenem-resistant *A. baumannii*, *P. aeruginosa*, and carbapenem-resistant Enterobacterales (CRE) were all classified as critical priority pathogens. These pathogens are particularly common in the Arab countries of the Middle East [14]. It is believed that the COVID-19 pandemic will lead to a worsening AMR pandemic; however, the full impact is yet to be entirely understood. A recent report from the American National Health Safety Network (NHSN) suggests a sharp increase in the rates of resistant pathogens post COVID-19 [15]. Since COVID-19 presents mostly with respiratory symptoms including fever that can be easily confused with bacterial infections, the spread of the virus was associated with a sharp increase in antimicrobial use [16,17]. In addition, a large percentage of hospitalized patients with COVID-19, particularly those requiring mechanical ventilation and intensive care unit (ICU) admission, received broad-spectrum antimicrobial agents despite negative cultures [17,18]. Because of the concern of acquiring infections with drug-resistant pathogens, broad-spectrum antibiotics were used for an extended duration in these patients.

Multi-faceted interventions are needed to halt the progression of bacterial resistance [6,19]. Globally, the United Nations (UN) and WHO were leading efforts to counter AMR before the COVID-19 pandemic [20] and urged individual countries and international organizations to develop and adopt strategies based on five pillars to combat AMR [19,21,22]: AMR awareness, AMR surveillance, infection prevention and control, optimization of antimicrobial use or antimicrobial stewardship (AMS), and sustainability through research [19]. In October 2015, the WHO launched the Global Antimicrobial Resistance and Use Surveillance System (GLASS) that was the first global collaborative effort to standardize AMR surveillance, and is an essential tool to inform infection control and prevention entities helping them to create policies [23]. It is the cornerstone for assessing the spread of AMR and monitoring the impact of local, national, and global strategies [23]. AMS includes interventions aimed at optimizing the judicious use of antibiotics to limit the spread of resistance.

The COVID-19 pandemic came, however, to divert such efforts. Following the COVID-19 pandemic, in May 2021, a report from the WHO showed that an increasing amount of data was being reported to GLASS, suggesting worrying trends, particularly in low- and middle-income countries (LMICs) [24]. Even though it may be early to link the increased reported resistance rates to the COVID-19 pandemic, the report included information on more than 3 million laboratory-confirmed bacterial infections caused by WHO priority list pathogens in 70 countries. There is a six-fold increase in resistance rates since sites first began reporting AMR surveillance data in 2017 [23,24]. Before the COVID-19 pandemic, the countries of the Arab world were already experiencing alarming rates of AMR [25]. Recognizing the limited published literature on the topic, our paper aims to assess the potential impact of COVID-19 on AMR and AMS in the countries of the Arab League. We also intend to discuss obstacles in implementing the rational use of antibiotics, monitoring resistance trends in the midst of the pandemic, and the effect on the economic crisis in some countries. Our objective is to raise awareness about the potential impact of antibiotic overuse during the pandemic and to suggest realistic approaches to combatting this threat.

## 2. AMR in the Arab World before COVID-19

The Arab world comprises countries in Africa and Western Asia that are grouped under the Arab league. Geopolitically, the Arab world is divided into three main regions: Gulf countries, Levant countries, and North African countries (Figure 1).

The Arab League is a regional organization in the Arab world, which is located in Africa and Western Asia. It can be geopolitically subdivided into three main regions, the Gulf countries, the Levant countries, and the African countries. The Gulf countries of the Arab League are the members of the Gulf Cooperation Council (GCC) (Bahrain, Kuwait, Oman, Qatar, Saudi Arabia, and United Arab Emirates (UAE)) and Yemen. The Levant countries of the Arab League are Iraq, Jordan, Lebanon, Palestinian territories, and Syria. The African countries of the Arab League are Algeria, Comoros, Djibouti, Egypt, Libya, Morocco, Mauritania, Somalia, Sudan, and Tunisia.

Several factors influence AMR surveillance and AMS implementation in Arab countries. This part of the world is a hotspot for infectious diseases and a cultural, religious, and ethnic mosaic of people. In addition, to cross border travel, these countries attract millions of international travelers for business, tourism, and pilgrimage. Those countries vary in resources, growth indices, and economic strengths. Major gaps exist between them, especially regarding healthcare expenditure and budgeting. The Gulf countries, for example, enjoy relative economic stability and wealth that facilitate funding of national and regional projects on AMR surveillance and AMS [9,10]. The Gulf Cooperation Council (GCC) established in 2005 the GCC-center for infection control to overcome AMR [26]. This center has created a strategic plan for improving AMR diagnosis and benchmarking. GCC countries have established national AMR surveillance plans in collaboration with the WHO [27]. In contrast, some of the Arab countries in North Africa and the Levant suffer from political instability and conflicts that have weakened their health infrastructures and displaced millions of people, hence hindering AMR surveillance [14,28,29].

From the pre-COVID-19 era, AMR figures were alarming in the Arab world. High proportions of carbapenem resistance in *Acinetobacter* spp. have been varying between 70 and 80% in the Levant, Gulf, and Egypt [30,31,32,33]. Among the Enterobacterales 3GCR organisms have reached more than 60% in *E. coli* and *Klebsiella* spp. [30,31,32,33] and carbapenem resistance has fluctuated between 10 to 30% [30,31,32,33]. *P. aeruginosa* resistant to carbapenems has been highly reported from some of the Levant and North African Arab countries (more than 50%), while reported to a lesser extent in the Gulf [14,33]. As to the molecular mechanisms of resistance, expression of carbapenemases like blaOXA-48 and blaNDM has been predominant among CRE, in addition to resistance mechanisms due to other β-lactamases, as well as outer membrane impermeability, and efflux pump [14,25,33]. Carbapenem resistance in *A. baumannii* in the region results from coexisting mechanisms. The main one is the expression of β-lactamase, particularly carbapenemases encoded by blaOXA-23, blaOXA-24, blaOXA-51, in addition to permeability and target modifications. The production of metallo-β-lactamases (mostly VIM and IMP) has been the most important mechanism of carbapenem resistance in *P. aeruginosa* throughout the Arab League countries [14,25,33]. Regarding Gram-positive bacteria, *Staphylococcus aureus* isolates have shown high methicillin resistance percentage of around 50%, while vancomycin resistance ranged between 0 and 5% [30,31,32]. In addition, penicillin non-susceptibility among *Streptococcus pneumoniae* has been frequently reported in the Middle East (50%) [14,32].

Several factors contribute to the increasing AMR in the region. A recent point prevalence study on infections in seven Arab countries suggested that 28.3% of the hospitalized patients have infections and 98.2% of them are receiving antibiotics [26]. Moreover, antimicrobials are available over the counter in most Arab countries and prescription regulation in the community setting is almost absent [34,35]. In addition, overuse of antibiotics in agriculture and animal feeds in some of the Arab countries has been correlated with increased resistance. Several studies from the region have shown high rates of resistance in agriculture and animals including 3GCR Enterobacterales, and more recently colistin-resistant pathogens [36]. In addition, in many Arab countries, particularly those in conflict zones, a large amount of poor generic antibiotics is available on the market. Some of the factors contributing to the rise in AMR include the lack of adequate laboratory facilities and comprehensive surveillance programs, limitations in infection prevention and control programs, and over-the-counter prescription of antibiotics including poor generic versions which can play a role in selecting for resistance [37]. The conflict in some of these countries has created a perfect storm for AMR exacerbated by armed conflicts and civil unrest. Studies have identified the nature of weapons used during conflicts and wars as a contributing factor to worsening AMR. Some field hospitals in Syria, Libya, and Iraq reported various wound contamination [32,38]. In Syria particularly, the use of conventional weapons, such as bombs, sea and land mines, and missiles, resulted in open wounds and fractures, amputations, as well as brain and/or spinal cord trauma. Due to the damage caused by war and collapsing healthcare systems, these injuries were contaminated at the time of trauma and possibly inappropriately treated, thus, creating a wide range of resistant microorganisms. An AMR surge was observed in Syria after the onset of several political and economic issues. Many of these problems were intensified by the limited number of trained medical professionals due to emigration and financial restrictions. These obstacles resulted in ill-equipped facilities, delays in reporting results, and poor microbiology infrastructures affecting result accuracy [38]. In addition, organisms commonly incriminated in wound infections such as *Acinetobacter baumannii* seemed to thrive in the presence of heavy metals in armed conflicts [39].

Due to this heterogeneity, AMR and AMS efforts in the Arab world have not reached the point of having a regional network. However, the WHO’s Regional Office for the Eastern Mediterranean (EMRO) has assisted almost all the countries in this region to develop and implement their national action plans to reduce AMR. Most Arab countries have produced AMR surveillance reports and many of them have reported their data to the GLASS [6,34]. Yet, the majority of the AMR data is generated by hospital laboratories and therefore may be more biased towards AMR in nosocomial organisms [30,31,32]. Several reports describe the AMS efforts in the Arab world [29,40,41] showing the successful implementation of stewardship programs to promote appropriate antibiotic use.

Efforts at curtailing the AMR in the region started several years ago and recently, we witnessed more concentrated efforts and collaborations among institutions and nations [28]. Though AMS efforts and results are limited, major efforts ensued to improve surveillance (as evidenced by the reporting through GLASS); and some countries in the GCC have had a good start with AMS (though results of AMS interventions are not published yet). Despite the absence of a pan-Arab AMS network, several initiatives on various fronts have culminated in establishing point prevalence surveys of infections, developing regional guidelines for the management of MDR infections [28], creating national action plans [29,30], strengthening regional networks and promoting AMS awareness and education among healthcare workers [42]. Experts from the Arab countries are also contributing to global AMS efforts as members of international organizations such as the WHO, the Alliance for Prudent Use of Antibiotics (APUA), the Surveillance and Epidemiology of Drug-Resistant Infections Consortium (SEDRIC), and other global networks [28].

## 3. The COVID-19 Pandemic: What Do We Know?

The COVID-19 pandemic has caused an unprecedented surge in hospital admissions, overwhelming healthcare systems in several cities across the world especially in critically ill patients [43]. The countries of the Arab world are no exception. SARS-CoV-2 pneumonitis can be easily confused with bacterial pneumonia. This translated into an increase in prescribing antimicrobials despite a low rate of bacterial co-infections and unclear clinical benefits for COVID-19 patients [44,45]. This overuse of antimicrobial agents is expected to significantly hinder AMS efforts, and to contribute to a rise in AMR globally [16,18,46,47]. The concern is that the most prescribed antibiotics including broad-spectrum antibiotics (carbapenems, tigecycline, cephalosporins, etc.) will become ineffective. In addition, during the pandemic, all available medical resources were repurposed to the COVID-19 response including reassigning tasks and duties of skilled AMS pharmacists and physicians. This indirectly resulted in decreased AMS activities [48].

### 3.1. Overuse of Antimicrobial Agents

The rise in antimicrobial usage was noticed across all categories of agents.

#### 3.1.1. Overuse of Antibiotics

The COVID-19 pandemic panic and the associated increase in critically ill patients led to an uncontrolled, frequently unjustified use of antimicrobials, in both critical care and outpatient settings. Azithromycin is one example: this agent was the second most prescribed medication for COVID-19, with 41% of healthcare providers reporting prescribing it for their patients [49]. In this study, 33% of COVID-19 patients were self-medicating with azithromycin before hospital admission [49]. In several instances, the increase in antimicrobial usage was not justified [50]. A review by Rawson et al. showed that 72% of hospitalized COVID-19 patients received antimicrobial therapy while only 8% had documented bacterial and/or fungal co-infection [51]. Similarly, a living meta-analysis by Langford et al. revealed that three-quarters of COVID-19 patients received antibiotics while the estimated prevalence of bacterial or secondary co-infection fluctuated between 6.1% and 8.0% [18]. Agents recommended for treating community-acquired pneumonia such as azithromycin, amoxicillin-clavulanate, levofloxacin, and amoxicillin were found to be among the most common inappropriately prescribed antibiotics in the United States, mainly due to unnecessary treatment of acute respiratory symptoms [52]. This suggested that COVID-19 has an immediate impact on outpatient antibiotic prescriptions. Other factors that could have contributed to the overconsumption of antibiotics included limited capacities of clinical factors and biomarkers to differentiate between viral and bacterial pneumonias. In addition, there are laboratory challenges in identifying co-pathogens in severe COVID-19 on time, as lower respiratory tract sampling is not recommended for fear of SARS-CoV-22 transmission during the procedure [53]. Data on antibiotic consumption during the COVID-19 pandemic from the Arab countries have not been published yet. However, we believe that the trends are not different from the published reports from western countries. On the contrary, with the lack of antimicrobial governance in many countries of the Arab world, the overuse of antibiotics was likely rampant.

#### 3.1.2. Overuse of Antivirals

Early in the pandemic, some antivirals were repurposed for treating SARS-CoV-2, including HIV protease inhibitors (lopinavir/ritonavir), nucleoside analog inhibitors of the RNA-dependent RNA polymerases (favipiravir), and neuraminidase inhibitors (oseltamivir) [54]. Despite the introduction of remdesivir as a potent SARS-CoV-2 replication inhibitor, physicians in several areas of the world continued using some available antivirals that targeted specific steps within the life cycle of SARS-CoV-2 [53]. Several countries in the Arab region have established national guidelines and protocols for the management of COVID-19 [55,56]. On another front, the reactivation of EBV and CMV led to over-treating with ganciclovir or other therapies. In one study, 85% of critically ill COVID-19 patients developed EBV (82%) or CMV (15%) [57]. Screening for those viruses could lead to an increase in the usage of ganciclovir and other therapies [58].

#### 3.1.3. Overuse of Antifungals

There is an increased interest in aspergillosis and other invasive fungal diseases in COVID-19 patients, specifically in ICU intubated patients [59,60]. COVID-19-associated pulmonary aspergillosis (CAPA) has a wide range of reported incidence and prevalence. Because of the difficulty in confirming the diagnosis, the European Confederation of Medical Mycology (ECMM) and the International Society for Human & Animal Mycology (ISHAM) established criteria for diagnosis and management [59,61]. The widespread use of systemic steroids for a prolonged time raises concerns about developing invasive candidiasis leading to abuse in voriconazole and other antifungals [62]. Outbreaks of *Candida auris* have been reported post-COVID-19 in many countries including four South American countries and Arab countries such as Lebanon where this fungus had never been isolated in the pre-COVID-19 era [63]. This is likely due to overconsumption of antimicrobial agents, particularly antifungal drugs, as well as poor infection control practices during the pandemic [63,64,65,66]. More recently, mucormycosis has emerged as an important and deadly co-infection in India and some countries of the Americas [67]. Although reports on mucormycosis were published from some countries of the Arab region in the pre-COVID-19 era [68], there are no new reports since the beginning of the pandemic.

#### 3.1.4. Widespread Use of Antiparasitic Agents

Chloroquine (CQ), hydroxychloroquine (HCQ), and ivermectin are still utilized worldwide despite the strong recommendation of the WHO against their use for the treatment of COVID-19 of any severity. This recommendation is mainly based on the lack of benefit seen in randomized clinical trials (RCTs) and the potential toxicity with those agents. Guideline development groups suggest that evidence on whether ivermectin reduces mortality, the need for mechanical ventilation, hospitalization, or time for clinical improvement is largely uncertain [69]. The potential impact on parasite resistance such as malaria in specific areas of the world is still unclear. Arab countries, particularly Egypt, have participated in some of the studies assessing the potential benefit of ivermectin [70,71]. Despite the lack of definite evidence, many physicians in the region continue to recommend ivermectin in the initial phase of COVID-19 management.

### 3.2. Shortage of Drugs

The pandemic disrupted the production and distribution of several medications. Lockdown measures led to airport/border closures and possibly to the repurposing of factories to produce COVID-19 therapies, which affected stocks of drugs around the world. An example of such was the significant shortage of antimalarial drugs and immune modulators [72]. Similarly, the shortages in diagnostic supplies and tools negatively influenced AMS efforts in timely de-escalation and switching from empiric to targeted therapies. Some countries in the Arab world witnessed a total economic collapse, particularly Syria and more recently Lebanon [73]. They have significantly suffered from the lack of availability of rapid diagnostic tools which are essential in stewardship efforts. For example, the American University of Beirut Medical Center, a major tertiary care center in the region had to outsource fungal and mycobacterial cultures, antigen tests, and others due to the unavailability of lab resources as well the exodus of the skilled laboratory staff resulting from the economic collapse.

### 3.3. Infection Control Measures

During the pandemic, several factors may have contributed to the transmission of resistant organisms, such as *C. auris*, and Acinetobacter species. Those factors included overcrowding of ICUs and shortage of personal protective equipment (PPE). These issues were caused by a high turnover of admitted patients, understaffing due to high rates of COVID-19 infections among healthcare workers, and quarantine measures [46]. In some academic medical centers in the region, the high number of infected staff followed the epidemiology of the COVID-19 infection nationwide [74]. Understaffing has been previously correlated with higher transmission of resistant pathogens as the staff tends to pay less attention to infection control practices when covering several patients during a shift [75].

### 3.4. Cleaning Products and Biocidal Agents in the Environment

Globally, another challenge imposed by COVID-19 is the impact of overuse of cleaning products in hospital settings and homes that end up in the sewers and the environment [76,77]. Low-level exposure to biocidal agents can select for drug-resistant strains and enhance the risk of cross-resistance to antibiotics, especially those used against Gram-negative organisms [47,78]. The excessive use of antibacterial products bearing biocides during the COVID-19 pandemic has disrupted the functioning of native microbes and exacerbated microorganisms resistance trends [78].

## 4. COVID-19 and AMR in the Arab World

Several factors contribute to the increase in AMR during the COVID-19 pandemic (Figure 2). Despite limited published data on the repercussions of COVID-19 on resistance rates in the Arab world, we expect the impact to be magnified as these countries face significant additional challenges compared to other regions of the world, as detailed earlier, and have limited inter-national collaborations in this field [28,32]. The impact of the political instability, geopolitical circumstances, armed conflicts, and economic challenges exacerbates the situation [32]. The published studies so far are relatively small, limited in scope, and not nationally representative. We recognize that there are no published reports directly comparing antimicrobial resistance rates in the pre-COVID-19 and post-COVID-19 eras from the Arab countries. There is a limited number of published reports on the issue. This may severely hinder the possibility of accurately determining the impact of the pandemic on AMR in the region. Nevertheless, the few published reports are alarming. For example, a report from Qatar described 98 isolated MDR Gram-negative bacteria from critically ill COVID-19 patients. The most frequently identified organisms were *Stenotrophomonas maltophilia* (24.5%) and *K. pneumoniae* (23.5%) [79]; 8.7% of *K. pneumoniae* isolates and 85.7% of *P. aeruginosa* isolates were found to be carbapenem-resistant [79]. In another study looking at COVID-19 patients in Egypt, 10.7% of the hospitalized patients showed bacterial and/or fungal co-infections. Drug resistance was commonly identified among bacterial pathogens, including *A. baumannii*, CRE and resistant *Pseudomonas* spp. [80]. It is highly desirable to have regional surveillance studies to accurately assess the impact of the COVID-19 pandemic on AMR in the region. We hope that the upcoming WHO-EMRO GLASS report would be highly informative. Outbreaks of clusters like *C. auris* were observed. A recent outbreak in Lebanon identified fourteen cases of *C. auris* infection/colonization [63], and by the time the manuscript was published, a total of 112 cases had been identified. All of the infected and colonized patients had multiple comorbidities and the majority were infected with SARS-CoV-2 before isolation of *C. auris* species [62].

The healthcare infrastructure in this region is heterogeneous. Yemen, Syria, and Libya face dismantled healthcare systems, whereas medical care is very developed in Lebanon and in the GCC countries [32] who benefit from successful Antimicrobial Stewardship Programs (ASP) and collaborations. On the other hand, there is no established ASP in the Palestine Authority, Syria, Libya, and Yemen [32,38]. In the vulnerable countries where healthcare systems are underdeveloped and because of the lack of AMS experts, we expect even greater overuse of antibiotics. A recent survey conducted by the Infectious Diseases Working Group in Arab countries of the Middle East highlighted the importance of promoting cross-regional collaboration in AMS [28]. The Working Group comprised experts in infectious diseases from Arab countries assembled to review similarities and differences in antimicrobial practices and management of multidrug-resistant organisms (MDROs) across the region and to assess the barriers to achieving cross-regional collaboration. An anonymous online survey showed heterogeneity between countries in awareness of local epidemiology, management of MDROs, and AMS practices. Key barriers to effective management of MDROs were discussed to guide the development of future coherent strategies to promote effective AMS in the region. Moreover, there is a great need to focus on training and education, capacity building, infrastructure, regional research, and regional surveillance. There are no restrictions on the use of over-the-counter antimicrobials [81] and at times, an antimicrobial prescription is almost unjustified especially in the treatment of COVID-19 with different classes of antimicrobials without proven efficacy. In addition to the economic crises and scarcity of resources, the problem is magnified by the production of low-quality generic medications, which are manufactured in plants that do not conform to standards. The sub-therapeutic dosing of the active ingredients can consequently lead to the selection of resistant strains [82,83].

Another rising challenge is the continued impact of the COVID-19 pandemic. The World Bank and the International Monetary Fund (IMF) announced that Arab countries are slow to implement nationally widespread COVID-19 vaccination campaigns [84,85]. This implies that most of the countries in the Arab world, except the GCC countries, will suffer from a sustained pandemic which will further weaken their healthcare systems and potentially contribute to increasing trends in AMR. A significant loss in gross domestic products is expected in LMICs due to AMR by the year 2050, which is going to be further reduced by the economic slowdown in the post-COVID-19 period [86]. These financial consequences may pose a further threat to the Tuberculosis (TB) programs that are already affected by the COVID-19 pandemic, leading to the potential spread of extensively drug-resistant TB in those countries [87].

With the increasing burden of the COVID-19 pandemic and the consequent AMR, we expect a surge of AMR in the Arab world and an increase in critically ill COVID-19 patients infected with MDROs which are associated with higher mortality rates.

## 5. Main AMS Interventions Obstacles and Proposed Solutions

AMS refers to coordinated interventions designed to measure and improve the appropriate use of antimicrobial agents by promoting the selection of the optimal antimicrobial drug regimen, including dosing, duration of therapy, and route of administration [88,89]. It applies to antibiotic use in the community, hospitals, long-term care facilities, as well as in the environment, veterinary, and agriculture fields [90,91]. In the Arab world, AMS practitioners and experts highlight the lack of enforcement of policies and guidelines by ministerial/governmental agencies [92]. According to regional experts, even though strategic plans exist, their implementation is hindered by weak governance: often no special unit at ministry level overseeing national AMS plans, no multidisciplinary coordination, no engagement from other sectors like environment and animal sectors, lack of political will due to poor understanding, etc. Other barriers to local and national ASP implementation and/or adoption include shortage of AMS experts, lack of education and training, poor communication, lack of specialized health information technology, in addition to the effect of war, violent conflicts, and migration [28,41,92]. Another important factor pertains to physicians’ fears and concerns concerning liability, and pressure from patients demanding antimicrobial prescriptions. Based on the discussion, we compiled in Table 1 the obstacles that hinder the implementation of comprehensive AMS programs at every step of the process.

The possible solutions needed in this area of the world to overcome the aforementioned obstacles include interventions such as strengthening AMR governance efforts at all levels, starting from the highest decision-making political level to the hospital administrative level reaching the individual practitioner. This can be achieved by prioritizing this issue in the ministries of health and the hospital organograms, and by creating independent entities on AMS and Infection Prevention and Control (IPC) with dedicated personnel and budget, along with clear road maps and key quality performance indicators [19,29]. At the healthcare facility and academic levels, improving targeted AMR education of healthcare workers in general is of utmost importance. Fighting AMR should be part of every healthcare worker’s culture [93]. Moreover, the development and application of treatment guidelines tailored to local epidemiology and AMR trends is crucial, where their dissemination and application would help reducing AMR emergence [94]. These efforts should go hand-in-hand with the reinforcement of IPC measures. Putting robust IPC plans across institutions at the country level would preventing the spread of difficult-to-treat organisms [19]. In addition, investing in the laboratory infrastructure required for the prompt identification of AMR and the detection of resistance mechanisms is a necessity. This laboratory capacity building is a real need in the countries of the Arab world [95]. A representative AMR surveillance that reflects the spread of resistance in the community and in hospitals would improve antimicrobial utilization and would limit the overuse of broad-spectrum antimicrobials [37,96,97].

Controlling the COVID-19 pandemic is of paramount importance to curb and slow the AMR tide. Vaccine stewardship is a new facet of AMS, whereby vaccine procurement by governments and well-established vaccination plans that take into consideration the availability of different vaccines are essential [98]. Different types of COVID-19 vaccines are offered in the countries of the Arab world with variable efficacy and effectiveness [99]. Addressing vaccine hesitancy that is hindering wide vaccine coverage in some countries of the region is another key issue [100]. Unfortunately, to date some countries under conflict continue to have poor or no access to COVID-19 vaccination. Strengthening AMS practices in COVID-19 patients through putting clear management guidelines of antimicrobials prescription and use of biomarkers such as procalcitonin might help decrease the impact of the pandemic on antimicrobial use [17,101]. Furthermore, clear national guidelines on antiviral and antiparasitic use in COVID-19 would also address the issue of overuse of these drugs. National efforts on the dissemination and application of these guidelines should be done to promote their proper implementation. Encouraging research and information exchange, as well as, putting national processes and analyzing outcome indicators would help keep track of AMS activities during the pandemic.

In addition, AMS should also involve collaboration with other sectors with the concept of the one health approach including the agriculture, animal husbandry and environmental use of antimicrobial agents. Some of the Arab countries have already initiated work in this area [27], however a regional approach is important particularly that meat products, fruits and vegetables from certain countries are sold in other countries and could serve as vehicles for transmission of resistant pathogens.

## 6. Conclusions

The COVID-19 pandemic has contributed to a worsening of the AMR pandemic globally, and the Arab countries are no exception. Several strategies are key in preparing for the future. Addressing the AMR with the One Health approach, involving sectors other than the human sector is important to control the threat of AMR. Enforcing infection control practices and encouraging immunization is important to curtail both pandemics. It is equally important to improve the preparedness of the public health sectors and primary care organizations with emphasis on training and support to deal with future pandemics and to strengthen regional and international AMR surveillance systems. The integration of rapid diagnostic tools will aid in sharpening clinical decision-making and impact stewardship efforts in treatment approaches. A budget to fund prevention and response efforts in the fields of AMR and AMS is highly needed to improve those efforts.

On the public health aspect, it is crucial to educate the wider public and to popularize AMR as a social concern. There is also a need to intensify political and popular support for efforts to tackle AMR across society, including the youth, and to draw the attention and action of political leaders, the private sector, and civil society.

Although it may be too late to build a unified response for all the Arab countries of the Middle East, it might not be too late to show some coordination—especially for richer states to provide logistical, technical, and financial assistance to their neighbors. The role of WHO EMRO has been instrumental in bridging gaps and supporting regional readiness. No country is isolated from the other, so cooperation and coordination are necessary to prevent a possible public health catastrophe.

## Figures and Tables

**Figure 1 antibiotics-10-01320-f001:**
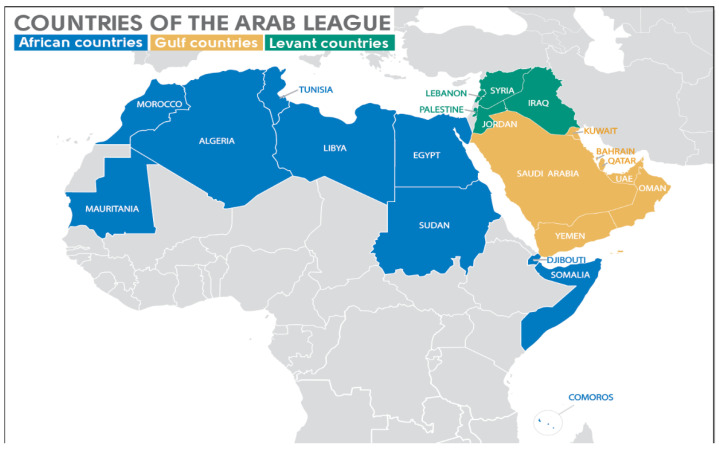
Map of the Arab League countries. The Gulf countries of the Arab League are the members of the Gulf Cooperation Council (Bahrain, Kuwait, Oman, Qatar, Saudi Arabia, and United Arab Emirates [UAE]) and Yemen. The Levant countries of the Arab League are Iraq, Jordan, Lebanon, Palestinian territories, and Syria. The African countries of the Arab League are Algeria, Comoros, Djibouti, Egypt, Libya, Morocco, Mauritania, Somalia, Sudan, and Tunisia.

**Figure 2 antibiotics-10-01320-f002:**
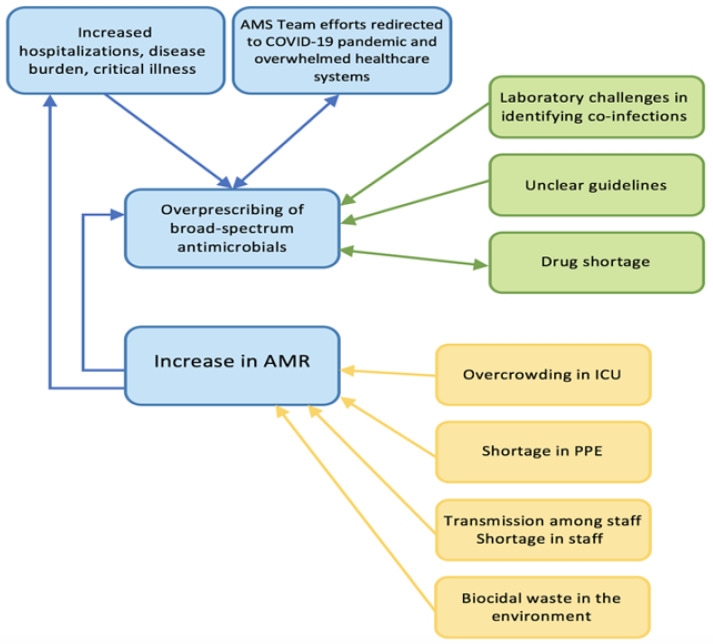
Factors contributing to the increase in AMR during the COVID-19 pandemic. AMR, antimicrobial-resistant; AMS, antimicrobial stewardship; ICU, intensive care unit; PPE, personal protective equipment.

**Table 1 antibiotics-10-01320-t001:** Main AMS interventions: possible obstacles. We list the most common AMS interventions used by Antimicrobial Stewardship Programs and the potential obstacles and limitations due to the COVID-19 pandemic and some related to regional peculiarities.

	Poor Quality of Generics	Sub-Optimal Laboratory Facilities *	Shortage of Antimicrobials	Lack of Education	Lack of Evidence in the Context of COVID-19	AMS Team Efforts Redirected to COVID-19
Use of broad-spectrum antimicrobials		X	X	X	X	X
Duration		X		X	X	X
De-escalation	X	X		X		X
Duplicate Therapy		X	X	X	X	X
Drug Bug Mismatch		X	X	X		X
Appropriate dosing regimen (drug choice, dosing)				X	X (DDI considerations)	X

IV, intravenous; PO, per os; DDI: drug drug interaction. * does not apply to all countries

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
