# Peer review of "Challenges to Antimicrobial Stewardship in the Countries of the Arab League: Concerns of Worsening Resistance during the COVID-19 Pandemic and Proposed Solutions"

_antibiotics, 2021, doi:10.3390/antibiotics10111320_

Round 1

Reviewer 1 Report

This is a paper demonstrating the impact of COVID-19 on antibiotic stewardship and resistance in the countries of the Arab league. I have some comments on your paper. Specifically,

If you submit it as a review, the reader expects the document to focus on evaluating interventions with a structured methodology (PRISM). Your paper is interesting but I think it is not the right journal to submit. For instance, the style is more appropriate for a book chapter as for a scientific journal.

I think there is too many uncertainties in your paper like "we believe that the trends are not different from the published reports from western countries" (l. 288) or " Despite limited published data on the repercussions of COVID-19 on resistance rates in the Arab world, we expect the impact to be magnified as these countries face significant additional challenges compared to other regions of the world" (l.306).

Please replace the term "abuse" of antibiotics, antivirals,... (l. 206, 231, 242) by overuse ? or inappropriateness ? 

Reviewer 2 Report

The aim of this manuscript is to study the potential impact of antibiotic abuse during the COVID-19 pandemic in the Arab world and to suggest approaches to counteract this threat. Overall, the manuscript is very interesting and well-written; still, there are some concerns that should be addressed.  

Bacteria names should be in Italic.

Line 151-2 – Sentence “In addition, in many of the Arab countries...” should be rephrased, or should be deleted because it lacks the verb needed for understanding.

Line 304 – Title of the section should be numbered 4., as the previous section is 3.The COVID-19 pandemic: what do we know?

Line 347-52 – Sentences about AMR surge in Syria do not belong to the section COVID-19 and AMR in the Arab world (This study was done before COVID-19 pandemics according to ref.). Thus, the authors have to decide to delete it or to incorporate it into the appropriate section.

Line 326 – ASP abbreviation should be explained.

Line 371-82 and Line 425-28 – There are no references cited to support claims.

Line 432 – In Table 1 there is no explanation of what DDI abbreviation stands for. Authors should put the explanation in Table’s legend below the title. In the same table, it is not clear what the ** symbol stands for (Sub-optimal laboratory facilities).

Reviewer 3 Report

The review by Rizk and colleagues focuses on AMR and AMS in the Arab world, also considering the impact of COVID-19.

Despite being of potential interest, the review suffers from several problems.

The authors state that it may be too early to determine the impact of the COVID-19 pandemic on AMR. This idea may be true in general, but it looks even more correct when considering the very limited number of cases of AMR infections observed over the pandemic in the Arab world (as reported in paragraph 3. COVID-19 and AMR in the Arab world”).

Second, the review misses a direct comparison of pre- and during COVID-19 AMR levels in the area of interest. Despite the numbers being reported for both periods, considering the heterogeneity of Countries (and hence health care settings and treatments, as also reported by the authors) and clinical status of the patients, a more precise comparison should be carried out to depict the actual situation. The central topic of the manuscript (according to the title and to the content of the introduction), the increase of AMR, is actually vaguely reported and limited to a single paragraph (L304-323). There are two possible solutions to these issues: either the authors delve into the comparison of the AMR levels before and during the COVID-19 pandemic, or they change the focus of the manuscript. For instance, a more appropriate title for the current version of the manuscript could be “A review on antimicrobial resistance and stewardship and proposed solutions in countries of the Arab league”.

Minor comments

Please write the bacterial species names in Italic.

L15: maybe better “worsen” than “deepen”?

L52: please correct “suggest” with “suggests”

L55: I believe “community” is not appropriate here. In fact, even hospital-acquired bacterial infections may have the same symptoms.

L110: I believe “ethnic” is more appropriate than “racial”

L151-152: please check the sentence, it looks truncated or incomplete.

The entire chapter 2 should be streamlined, as it is not

Round 2

Reviewer 1 Report

ok for me.  I better understand the context. 

Author Response

Thank you very much for your positive feedback.

Reviewer 3 Report

I am afraid the authors did not meet my comments of major concern.

In particular, they have changed the title, but still referring to the COVID-19 pandemic situation. Yet, they have not properly contextualized nor quantified the supposed impact of the pandemic on the AMR.

Secondly, when I criticized the fact that the number of cases under analysis  may severely hinder the possibility of determining the impact of the pandemic on AMR, I expected that the authors would have included some notes and comments on this in the manuscript. And yet, I could not find them in the new version of the manuscript.

Author Response

REVIEWER 3

I am afraid the authors did not meet my comments of major concern.

Thank you very much for your constructive comments which we highly appreciate.

In particular, they have changed the title, but still referring to the COVID-19 pandemic situation. Yet, they have not properly contextualized nor quantified the supposed impact of the pandemic on the AMR.

We apologize if we did not make our reply clear enough.

We have modified the Introduction to add more contextualization to the section to now read as such:

Before the COVID-19 pandemic, the countries of the Arab world were already experiencing alarming rates of AMR [25]. Recognizing the limited published literature on the topic, our paper aims to assess the potential impact of COVID-19 on AMR and AMS in the countries of the Arab League. We also intend to discuss obstacles in implementing the rational use of antibiotics, monitoring resistance trends in the midst of the pandemic, and the effect on the economic crisis in some countries. Our objective is to raise awareness about the potential impact of antibiotic overuse during the pandemic and to suggest realistic approaches to combatting this threat.”

In addition to changing the title of the manuscript, we tried to present all the available published literature on the effect of the COVID-19 pandemic on AMR in the Arab countries and added the following statement in Section 4 line 347 to now read as:

It is highly desirable to have regional surveillance studies to accurately assess the impact of the COVID-19 pandemic on AMR in the region. We hope that the upcoming WHO-EMRO GLASS report would be highly informative.

Secondly, when I criticized the fact that the number of cases under analysis may severely hinder the possibility of determining the impact of the pandemic on AMR, I expected that the authors would have included some notes and comments on this in the manuscript. And yet, I could not find them in the new version of the manuscript.

Thank you again for this valuable comment which allowed us to explain further the current situation. We have reworded Section 4 in particular the line 333 to add the following sentence including your suggested statements:  

We recognize that there are no published reports directly comparing antimicrobial resistance rates in the pre-COVID19 and post-COVID-19 eras from the Arab countries. There is a limited number of published reports on the issue. This may severely hinder the possibility of determining accurately the impact of the pandemic on AMR in the region. Nevertheless, the few published reports are alarming.”  

This manuscript is a resubmission of an earlier submission. The following is a list of the peer review reports and author responses from that submission.